# Dichotomous Regulation of Acquired Immunity by Innate Lymphoid Cells

**DOI:** 10.3390/cells9051193

**Published:** 2020-05-11

**Authors:** Takashi Ebihara

**Affiliations:** Department of Medical Biology, Akita University Graduate School of Medicine Affiliation, 1-1-1 Hondo, Akita 010-8543, Japan; tebihara@med.akita-u.ac.jp; Tel.: +81-18-884-6080

**Keywords:** innate lymphoid cells, NK cells, ILC1, ILC2, exhausted-like ILC2, ILC3, IL-10

## Abstract

The concept of innate lymphoid cells (ILCs) includes both conventional natural killer (NK) cells and helper ILCs, which resemble CD8^+^ killer T cells and CD4^+^ helper T cells in acquired immunity, respectively. Conventional NK cells are migratory cytotoxic cells that find tumor cells or cells infected with microbes. Helper ILCs are localized at peripheral tissue and are responsible for innate helper-cytokine production. Helper ILCs are classified into three subpopulations: T_H_1-like ILC1s, T_H_2-like ILC2s, and T_H_17/T_H_22-like ILC3s. Because of the functional similarities between ILCs and T cells, ILCs can serve as an innate component that augments each corresponding type of acquired immunity. However, the physiological functions of ILCs are more plastic and complicated than expected and are affected by environmental cues and types of inflammation. Here, we review recent advances in understanding the interaction between ILCs and acquired immunity, including T- and B-cell responses at various conditions. Immune suppressive activities by ILCs in particular are discussed in comparison to their immune stimulatory effects to gain precise knowledge of ILC biology and the physiological relevance of ILCs in human diseases.

## 1. Innate Lymphoid Cells

Innate lymphoid cells (ILCs) are a series of lymphocytes that are equipped with the ability to produce innate helper cytokines at the tissue or with innate cytotoxicity [1]. ILCs do not express antigen-specific receptors that require RAG1 or RAG2 recombinase. Instead, ILC differentiation is dependent on IL2γg and cognate cytokines, including IL-7 and IL-15 [2,3,4,5,6]. Recently, ILC subsets were classified into five groups: natural killer (NK) cell, ILC1s, ILC2s, ILC3s, and lymphoid tissue-inducer (Lti) cells. The concept of ILCs mirrors that of acquired cellular immunity composed of CD8^+^ killer T cells and CD4^+^ helper T cells (Figure 1) [1]. NK cells are innate counterparts of CD8^+^ killer T cells, and they are designated as cytotoxic ILCs that can kill target cells by detecting loss of major histocompatibility complex (MHC) Class I or activating ligands on the target cells [7]. Other ILCs are helper ILCs that produce helper cytokines like CD4^+^ helper T cells do. Helper CD4^+^ T cells are functionally divided into several subpopulations, as determined by the types of helper cytokines they produce. Helper ILCs are similarly classified into T_H_1 cytokine-producing ILC1s, T_H_2 cytokine-producing ILC2s, and T_H_17/T_H_22 cytokine-producing ILC3s [1]. Lti cells contribute to second lymphoid organogenesis with lymphoid tissue-organizer cells in embryos [8]. Helper innate lymphoid cells resembling Treg and Tfh cells have not yet been identified. Id3^+^ regulatory ILCs in murine intestine were reported to be possible innate counterparts of Treg cells because they produce an immunosuppressive cytokine, IL-10 [9]. However, other researchers reported that the IL-10-producing ILCs in the intestine were ILC2s [10].

The majority of helper ILCs are resident and self-renewed in tissue to achieve innate responses against harmful events [11,12], though some intestinal ILC2s and ILC3s can migrate to lymph nodes or other tissue types [13,14]. In contrast, NK cells move around the body to find aberrant target cells [11]. NK cells and ILC1s are very similar populations because both populations express T-bet and produce IFN-γ (Figure 1) [1]. Although NK cells require Eomes for their high cytotoxicity, ILC1s are generally negative for Eomes, with some exceptions in humans and mice [15,16,17]. Their dependency on T-bet is different between NK cells and ILC1s in mice. T-bet deletion leads to almost absence of ILC1s but not NK cells in murine liver [18]. Splenic NK cells, deficient in T-bet, exhibit defects in maturation, IFN-γ production, and cytotoxicity [19]. IL-15 and its receptor complex IL2Rβ/IL2rγ are necessary for NK-cell and ILC1 differentiation in mice [6]. Although IL7Rα is preferentially expressed in ILC1s like other helper ILCs, the genetic deletion of IL-7 does not reduce ILC1s in murine intestine [6]. A transcription factor, Hobit, is necessary for ILC1 differentiation, especially in mouse liver [20]. ILC2s are characterized by a high expression of GATA-3 and the production of T_H_2 cytokines such as IL-4, IL-5, and IL-13 (Figure 1) [1]. Murine ILC2s also produce IL-9 for their expansion after activation [21]. ILC3s have two subpopulations, natural cytotoxicity receptor (NCR)^+^ ILC3s and NCR^–^ ILC3s (Lti-like cells), both of which require RORγt and AHR for their differentiation and IL-17/IL-22 production (Figure 1) [1]. The differentiation of murine NCR^+^ ILC3s is also dependent on T-bet [22]. Lti-like cells are marked by CCR6 expression and produce more IL-17 and less IL-22 than NCR^+^ ILC3s in mice [23]. Thereby, NCR^+^ ILC3s and Lti-like cells are recognized as innate equivalents of T_H_17 and T_H_22 cells, respectively. Lti-like cells express MHC Class II and present antigens to CD4^+^ T cells [24,25], as discussed later. Lti cells in embryos resemble ILC3s because they are dependent on RORγt and produce IL-17/IL-22 [1]. IL-7 and its receptor complex IL-7Rα/IL-2rγ are indispensable for ILC2 and ILC3 differentiation in mice [3,4,5,6].

ILC differentiation and function is regulated by the Runx transcription factor family comprising Runx1, Runx2, Runx3, and their biding partner Cbfβ [26,27]. Heterodimer formation with Cbfβ is required for each Runx to bind genomic DNA; thereby, deletion of *Cbfb* abrogates all Runx protein function. Runx3 is differentially expressed by ILC subsets: Runx3^Hi^ ILC1s, Runx3^intermed^ ILC3s, and Runx3^Lo^ ILC2s [26]. Runx3 is essential for ILC1 survival and RORγt expression by ILC3s; depletion of Runx3 results in impaired ILC1 and ILC3 differentiation but not ILC2 [26,28]. Intermediate expression of Runx1 seems to compensate for the loss of Runx3 in ILC2s. Helper T-cell differentiation is also regulated by Runx proteins [29]. Runx3 is critical for CD8^+^ T-cell and T_H_1-cell differentiation and their effector functions [30,31,32]. Runx1 directly induces RORγt, which is a master regulator of T_H_17 and T_H_22 cells [33,34]. These data indicate that Runx proteins control helper responses in innate and acquired immunity.

ILCs can be found in almost every organ and tissue type, such as meninge, peripheral blood, skin, lung, liver, stomach, intestine, islet, adipose tissue, spleen, and lymph nodes [1,3,4,5,11,14,35,36,37,38,39,40,41]. However, mouse studies showed that the distribution of ILCs is quite variable [36]. Lungs are enriched in ILC2s and NK cells. ILC1 and NK cells are major ILCs in the liver. The intestine is armed with NK cells, ILC1, ILC2, and ILC3s. ILC3s are preferentially localized in mucosal tissue, such as the skin and intestine, where microbiota live close [5,42,43,44]. ILC3 fitness is affected by commensal bacteria. Such broad distribution of ILCs constitutes a global innate immune network. Originally, the physiological relevance of ILCs was investigated using RAG1- or RAG2-deficient mice lacking acquired immunity to observe robust effects. In the past few years, cumulative studies have demonstrated that ILCs clearly have immune-stimulatory and anti-inflammatory activities against acquired immunity. Some review papers summarized data regarding how ILCs modulate T cells and B cells [45,46,47]. However, a comprehensive review to clearly dissect ILC biology in the context of immune activation and suppression has not been published yet. Therefore, here, we focus on the functional dichotomy in ILCs including NK cells to positively or negatively regulate acquired immunity in various physiological and pathological conditions.

## 2. NK Cells, ILC1s, and Acquired Immunity

### 2.1. NK Cells and ILC1s Enhance Type I Immune Responses

NK cells and ILC1s are innate components of Type I immunity which provides protective responses against tumor cells or intracellular microbes, such as viruses, bacteria, and protozoa (Figure 2a). NK cells and ILC1s can be activated by cytokines or via direct contact with other cells expressing activating ligands [7]. NK cells express a series of activating and inhibitory receptors, both of which determine NK-cell activity through their interaction with ligands. For example, NKG2D is the most studied NK cell-activating receptor, of which the ligands are expressed on virus-infected cells and tumor cells [48,49]. Direct contact with these cells activates NK cells. Other activating receptors include CD16, NCRs (NKp46, NKp44, NKp30), DNAM-1, and CD27 in humans and mice [48,50,51,52,53,54]. Major NK-cell inhibitory receptors are Ly49s in mice and KIRs in humans. MHC Class I on the target cells binds to Ly49s or KIRs and induces inhibitory signals in NK cells [7]. Another important NK cell receptor is CD94, which forms an inhibitory heterodimer with NKGA, or an activating heterodimer with NKG2C or E [55]. CD94/NKG2 receptors recognize non-classical MHC Class I: Qa-1 in mouse and HLA-E in human. NK cells do not attack the healthy cells normally expressing the self MHC Class I. Loss of the self MHC Class I on transformed cells provokes NK-cell activation due to the loss of inhibitory signals.

NK cells can be activated by conventional dendritic cells (cDCs) and plasmacytoid DCs (pDCs), both of which can recognize pathogen-associated pattern molecules (PAMPs) by Toll-like receptors (TLRs) and RIG-I-like receptors (RLRs) [56,57,58,59,60]. The activated cDCs produce IFN-α/β, IL-12, IL-15, and IL-18 or directly interact with NK cells for NK-cell activation [57,61,62]. NKG2D, NKp46, DNAM-1, and INAM on NK cells are involved in contact-dependent NK-cell activation by cDCs [7,63,64,65,66,67]. IL-15 trans-presented by IL-15 receptor α-chain (IL-15Rα) on cDCs is also critical for NK-cell activation through direct contact [68]. Although pDCs can be major IFN-β-producing immune cells in the spleen after a viral infection, ablation of pDCs minimally affects NK cell activity during mouse cytomegalovirus (MCMV) infection [69,70]. NK cells in turn promote DC maturation for efficient T cell priming by secreting GM-CSF, TNF-α, and IFN-γ [57,58,71].

NK cells are not only effector cytotoxic cells against virally infected cells or tumor cells, but also innate helper cells producing IFN-γ that promotes T_H_1 differentiation leading to cytotoxic T lymphocyte (CTL) activation (Figure 2a). When lipopolysaccharide (LPS)-stimulated dendritic cells (DCs) were subcutaneously injected into mice, DCs in the lymph node recruited NK cells in a CXCR3-dependent manner [72]. The recruited NK cells produced IFN-γ to promote T_H_1 differentiation. The deletion of NK cells by neutralizing antibodies or the genetic ablation of *Ifng* abrogated T_H_1 polarization, suggesting that NK cells work as a bridge between DCs sensing PAMPs and T_H_1-cell induction. This bridging function is utilized for development of new efficacious vaccine adjuvants. A combined adjuvant AS01 comprising a TLR4 ligand, 3-O-desacyl-4′-monophosphoryl lipid A, and a saponin, QS-21, can promotes T_H_1 polarization which is mediated by adjuvant-activated DCs and early IFN-γ secretion from NK cells [73]. In addition to IFN-γ, direct contact of NK cells with CD4^+^ T cells may be important for NK cell-mediated T cell activation. CD16-activated human NK cells upregulated OX40 which stimulated autologous CD4^+^ T cells in vitro [74]. However, in these experiments, the activated NK cells were fixed by paraformaldehyde before co-culture, thereby possible NK cell cytotoxic effects on activated T cells were not considered.

In the same line as NK cells, ILC1s are also activated by IL-12, -15, and -18 secreted by myeloid cells [2]. While ILC1s are known to express activating receptors such as NKG2D, NKp46, and DNAM-1, a limited number of Ly49 receptors are expressed in murine hepatic ILC1s compared with splenic NK cells [11,17,75]. Murine hepatic ILC1s can be activated by hepatocyte-expressing CD155, a ligand for DNAM-1, which is induced by chemical hepatic injury [75]. Thus, similar cytokine stimulation is required for ILC1 and NK-cell activation, though the mechanisms by which other immune cells or transformed cells activate ILC1s are poorly understood.

The physiological functions of hepatic ILC1s were studied in the MCMV infection model [76]. MCMV has been commonly used to study the biology of NK cells because MCMV encodes m157 that activates murine NK cells through activating receptor Ly49H in C57BL/6 mice [77]. MCMV m12 can also interact with NK1.1 on NK cells and ILC1s and activate them [78]. Therefore, NK cells are central immune cells for host defense against MCMV in C57BL/6 mice. When C57BL/6 mice are infected with MCMV through the peritoneal cavity, ILC1s in the peritoneal cavity are main IFN-γ producers until 24 h post infection [76]. Then, MCMV reaches the spleen and liver for their propagation. Increased viral burden in these organs was observed in the Zfp683 (encoding Hobit)^-/-^ mice that lack hepatic ILC1s. These data indicated that ILC1s limit MCMV infection by the early production of IFN-γ in mice. In addition, hepatic ILC1s persisted after acute MCMV infection and responded vigorously against a second challenge of MCMV infection [78]. However, in these studies, acquired immunity was not investigated because NK cells are critical for host defense against acute MCMV infection in C57BL/6 mice. Therefore, it remains elusive how ILC1s instruct acquired immunity to protect the host from viral infection. Although Zfp68^−/−^ mice are often used to study ILC1-specific biology, T cells are also affected in those mice [20]. Therefore, specific depletion of ILC1s, but neither NK cells nor T cells, is mandatory to study a link between ILC1s and acquired immunity.

### 2.2. Regulatory Functions of NK Cells and ILC1s

NK cells are appreciated as innate guardians against virally infected cells or transformed cells. However, the physiological roles of NK cells in antiviral immunity are more complex than expected. NK cells have apparently regulatory effects on acquired cellular and humoral immunity during viral infection (Figure 2b) [79,80,81,82,83,84,85]. Surprising evidence is that the depletion of NK cells saves mice from the development of chronic lymphocytic choriomeningitis mammarenavirus (LCMV) infection [79]. LCMV does not encode viral proteins that strongly activate NK cells like MCMV m157. NK-cell depletion increases T_H_1 cells and antigen-specific CTLs in mice infected with LCMV, suggesting that activated NK cells have regulatory functions against T_H_1 cells and CTLs [79,86]. The direct killing of CD4^+^ and CD8^+^ T cells by NK cells is suggested because NK cells eliminate transferred CD4^+^ and CD8^+^ T cells in mice [79,86,87]. NKG2D and 2B4 on NK cells are crucial for the killing [87,88]. Another possible mechanism of CTL reduction is the cytotoxic activity of NK cells against infected DCs resulting in limited exposure of viral antigens to CTLs, as observed in MCMV infection [89]. However, when mice are infected with a high dose of LCMV, NK cells, in turn, protect the host from a lethal infection [79]. A high dose of LCMV exhausts LCMV-specific T cells and causes persistent infection. NK cells suppress the exhaustion of CD8^+^ T cells through the elimination of activated CD4^+^ T cells. Thus, magnitude of virus infections influences NK cell behavior against T cell responses.

Humoral immunity is also regulated by NK cells. Depletion of NK cells at infection promotes Tfh-cell differentiation, resulting in a high neutralizing antibody titer against LCMV [90]. Recent mouse data indicate that perforin-dependent NK cell function contributes to reduced numbers of Tfh cells and B cells with high affinity maturation [84]. Therefore, NK cells repress proper immunoglobulin production by controlling development of Tfh cells.

Human NK cells have been shown to exert regulatory effects on CD4^+^ and CD8^+^ T cell via their cytotoxic ability. NK cell killing of activated autologous CD4^+^ T cells is mediated by the integrated signal from NKG2D, LFA-1, and TRAIL in vitro [91]. NKG2D ligands are induced on antigen-activated CD4^+^ and CD8^+^ T cells and allow autologous NK cells to eliminate these T cells in an NKG2D-dependent manner in vitro [92]. Immune suppressive roles of NK cells are implicated in human diseases. Antigen-specific CD8^+^ T cells in patients with chronic hepatitis B virus infection express a death receptor for TRAIL and are susceptible to NK cell killing [80]. Dysregulation of NK cell is associated with increased Tfh cells and development of broadly neutralizing antibody against HIV in patients [81]. Cytomegalovirus (CMV) infection increases frequencies of NK cells expressing an activating receptor NKG2C [93]. These NKG2C^+^ NK cells persist and expand well during CMV reactivation. HLA-E, a NKG2C ligand, is induced on CMV-specific CD8^+^ T cells which are preferentially eradicated by the NKG2C^+^ NK cells [82]. These data suggest that NK cells can be negative regulators of acquired immunity in humans as well as in mice.

NK-cell function is tuned by environmental cues, such as tumor microenvironments and chronic inflammation (Figure 2b) [94]. Dysfunctional NK cells are found in many tumors and chronic infections. Intratumoral NK cells show diminished cytotoxicity and IFN-γ production in mice and patients with various cancers [94,95,96]. In many cases, NK cell-activating receptors, including NKG2D, NCRs, and CD16, are reduced in the dysfunctional NK cells [94,97]. Checkpoint inhibitory receptors, such as PD-1, Tim-3, and TIGIT, are also upregulated and contribute to NK-cell dysfunction [98,99,100,101]. Among these inhibitory receptors, TIGIT on NK cells can mainly contribute to NK cell exhaustion in tumor-bearing mice [101]. Blockade of TIGIT unleashes anti-tumor CD8^+^ T cell by reversing mouse NK-cell dysfunction. An inhibitory NK-cell receptor NKG2A is expressed in half of peripheral NK cells and a small fraction of CD8^+^ T cells, and was found to be another checkpoint inhibitor that suppresses NK-cell effector functions against tumor cells expressing mouse NKG2A ligand, Qa-1 [102]. Efficacies of humanized NKG2A blocking antibody is being examined in clinical trials to augment antitumor cytotoxicity by NK cells and CD8^+^ T cells. Thus, NK-cell exhaustion can have a negative impact on CD8^+^ T immune responses against tumor.

However, NKG2A functions in NK cells and T cells are complicated. NKG2A on NK cells prevents NK cell killing of Qa-1-expressing pathogenic CD4^+^ T cells in experimental autoimmune encephalomyelitis, suggesting that inhibition of NKG2A-Qa-1 interaction suppresses CD4^+^ T cell responses to autoantigens [103]. On the contrary, genetic ablation of *NKG2A* enhances activation of CD8^+^ T cell during virus infections. NKG2A^-/-^ mice are more succumb to adenovirus and influenza infections than WT mice [104]. This may be attributable to exaggerated production of inflammatory cytokines by NKG2A^−/−^CD8^+^ T cells. However, CD8^+^ T cells can be overactivated and apoptotic in the NKG2A^-/-^ mice infected with ectromelia virus [105]. The apoptosis of the CD8^+^ T cells is cell-intrinsic and independent of NK cells. Thus, blockade of NKG2A might need cautions during virus infections.

NK cells activated by IL-12 or IL-15 can produce an immunosuppressive cytokine, IL-10, which suppresses acquired cellular responses in mice and humans (Figure 2b) [106,107,108]. NK cells are a major source of IL-10 following infections such as *Toxoplasma gondii*, *Listeria monocytogenes*, LCMV, and MCMV [106,108,109,110,111]. The genetic deletion of *Il10* causes body-weight loss accompanied by increased IFN-γ^+^ CD4^+^ and CD8^+^ T cells leading to a reduced viral burden in mice infected with MCMV [110,111]. However, the regulatory roles of IL-10 from NK cells during infections have been discussed for many years, because other immune cells can produce IL-10 during infections [109]. Mouse studies showed that NK-cell-specific depletion of *Il10* led to a comparable viral burden and T-cell responses during MCMV or LCMV infection [109,112]. Therefore, IL-10 from NK cells might not play a major role in regulating acute antiviral immune responses. However, IL-10 production by NK cells has been shown to intervene therapeutic effects of IL-15 complex (IL-15C) treatment in the murine model of experimental cerebral malaria [113]. IL-15C treatment is quite effective to prevent cerebral malaria. The advantageous effects are attributed to IL-10 mainly from NK cells and decreased CD8^+^ T cell activation. Furthermore, adoptive transfer of IL-15C-treated NK cells, but not IL-10^-/-^ NK cells, can recapitulate the therapeutic effect of IL-15C treatment. Thus, IL-10 production by IL-15C-activated NK cells dampens pathogenic CD8^+^ T cell responses against malaria.

Regulatory function of hepatic ILC1s on anti-viral T cells has been recently reported [114]. Hepatic ILC1s express programed death ligand 1 (PD-L1) which can interact with a checkpoint inhibitory receptor, PD-1, on hepatic T cells and suppress CD4^+^ and CD8^+^ T cell responses against LCMV. Therefore, murine ILC1s as well as NK cells can suppress antiviral cellular immunity and exhibit a dichotomy in modulating acquired immunity.

## 3. ILC2s and Acquired Immunity

### 3.1. ILC2s Enhance Type II Immune Responses

ILC2s produce T_H_2 cytokines at peripheral tissue and lymph nodes upon activation and initiate Type II immune responses that cause allergies or protect the host from parasitic worm infections (Figure 3a) [1,3,4]. ILC2s are the main producers of IL-5 that recruit eosinophils [115]. Amphiregulin and IL-13 from activated ILC2s collectively elicit mucus production, smooth muscle hypercontractility, and tissue repair [116,117]. Increased ILC2 activity and numbers were observed in patients with asthma, atopic dermatitis, and chronic rhinosinusitis compared to healthy controls [118,119,120], suggesting that ILC2s contribute to allergy pathogenesis. Most allergens have proteinase activity that can disrupt mucosal tissue. ILC2s are activated by allermins, including IL-25, TSLP, and IL-33, released from damaged epithelial cells and stromal cells [3,38,40,121,122,123]. Inversely, IFN-α, IFN-γ, and IL-27 suppress and terminate ILC2 activation during allergic inflammation [124,125]. ILC2 activity is regulated by secreted factors from neuron. Neuropeptides such as VIP and NMU23 activate ILC2s [115,126,127,128], while another neuropeptide, CGRP, constrains ILC2 proliferation and IL-13 production [129]. NMU23 is released from neuron and augments ILC2 activity through its receptor, NMUR1, during allergen-induced inflammation in the lung or helminth infection in the intestine [126,127,128]. Pulmonary neuroendocrine cells (PNECs) are sensory lung epithelial cells which produce CGRP. Ablation of PNECs in mice decreased allergic inflammation in the lung [130]. However, three papers indicated that ILC2s produced CGRP by themselves and inhibited ILC2 proliferation and IL-13 production, but not IL-5 production, in allergen-sensitized mice [129,131,132]. A neurotransmitter, β2 adrenaline, negatively regulates ILC2s activity through its receptor, β2AR, in mice infected with helminth [133]. Lipid mediators also modulate ILC2 functions. PGD2 and leukotrienes from mast cells and eosinophils have positive effects on ILC2 activity, though PGE2 and PGI2 inhibit ILC2 activity in humans and mice [134]. Activated ILC2s express ICOS and its ligand ICOSL. The ICOS/ICOSL interaction among ILC2s themselves supports ILC2 activity during their activation process [135]. Glucocorticoid-induced tumor necrosis factor receptor (GITR) expressed on ILC2s is also involved in ILC2 activation [136]. Thus, many soluble factors and receptors were found to be involved in ILC2 activation status.

Previous studies clearly showed that ILC2s promoted T_H_2-cell differentiation (Figure 3a). In the absence of ILC2s, T_H_2-cell response is impaired in mice challenged with an allergen or infected with parasitic worms [137,138]. Several mechanisms of ILC2-mediated T_H_2-cell induction have been reported. First, IL-13 secreted by activated ILC2s stimulates CD11b^+^CD103^−^ DCs to produce T_H_2-attracting CCL17 [139]. CCL17-producing DCs are required for memory T_H_2-cell recall response. Second, IL-13 from activated ILC2s also induce M2-type macrophages, which enhance T_H_2 differentiation [140,141]. Third, ILC2s express MHC-Class II and costimulatory molecules and present antigens to naïve CD4^+^ T cells for T_H_2-cell differentiation [142]. Direct antigen presentation is the ability that can be uniquely observed in ILC2s and ILC3s, but not helper T cells. ILC2s elicit antigen-dependent T_H_2-cell proliferation that is dampened by MHC Class II deficiency on ILC2s in humans and mice. Lastly, costimulatory molecule OX40L on ILC2s promotes T_H_2-cell proliferation through OX40 [143]. In this case, Treg-cell expansion is also induced through OX40/OX40L interaction, suggesting that ILC2s can increase pathogenic T_H_2 cells and immunosuppressive Treg cells in the mice treated with the allergen. However, ablation of OX40L in ILC2s results in reduced allergic inflammation; thereby, OX40L on ILC2s plays an important role in allergy pathology rather than immunosuppression. These findings indicated that ILC2s indirectly or directly enhance Type II immune responses by T_H_2-cell activation.

ILC2s also support humoral immunity that includes T-cell-independent innate B1-cell response and T-cell-dependent acquired B2-cell response (Figure 3a). B1 cells are abundant in peritoneal and pleural cavities and secrete natural serum antibodies. IL-5 produced by ILC2s in the fat-associated cluster of the mesentery promote the self-renewal of B1 cells but not B2 cells and enhance IgA production in C57BL/6 mice [3]. Another study observed that ILC2s elicited B2-cell proliferation in BALB/C mice, though the effects were not as pronounced as those in B1-cell proliferation [144]. Antihelminth IgE production is impaired in mice where OX40L is specifically deleted in ILCs, including ILC2s [143]. OX40L/OX40 signaling is known to be crucial for the interaction between DCs and Tfh cells [145]. Therefore, OX40L on ILC2s might directly stimulate OX40 on Tfh cells for IgE production in mice infected with helminth. Thus, ILC2s are competent to activate innate and acquired humoral immunity.

### 3.2. Regulatory Functions of ILC2s

ILC2s can promote Treg-cell expansion and suppress excessive immune responses (Figure 3b). As described above, Treg-cell proliferation is induced by direct contact with ILC2s through OX40/OX40L signaling [143]. ICOSL on ILC2s also stimulates ICOS on Treg cells for Treg-cell proliferation [146]. In a mouse model of arthritis, IL-9-producing ILC2s play a crucial role in the activation of Treg cells in mice and humans [147]. ILC2s progressively secrete IL-9 to keep the activity in joint fluid. The activated ILC2s express ICOSL and GITRL to support Treg-cell expansion through ICOS and GITR on Treg cells. Increased numbers of IL-9^+^ ILC2s were observed in the joints and peripheral blood of patients with rheumatoid arthritis in remission compared to those with acute arthritis. Therefore, IL-9-mediated ILC2 activation is essential for the resolution process by Treg cells in arthritis.

Immunoregulatory roles of IL-10-producing ILC2s have been suggested (Figure 3b). We and others exhibited that severe or repeated allergic inflammation induced IL-10-producing ILC2s [10,148,149]. IL-10 exerts immunosuppressive effects on dendritic cells, macrophages, T_H_2 cells, and stimulatory effects on Treg cells [150]. Seehus et al. first reported that IL-33 administration induced IL-10 production in murine lung ILC2s [148]. Stimulation with IL-2, IL-7, and IL-33 with retinoic acid (RA) strongly enhanced IL-10 secretion by naïve ILC2s. IL-10 production by ILC2s was correlated with reduced eosinophil recruitment. We also showed that in the mouse asthma model by papain nasal administration, high doses of papain treatments were required for the emergence of IL-10-producing ILC2s [149]. It took seven days to observe IL-10 production in ILC2s when mice were treated with papain every three days. Another group identified IL-10-producing ILC2s in the intestine [10]. A small percentage of intestinal ILC2s produce IL-10, of which the expression was induced in vitro by IL-2, IL-4, IL-27, IL-10, and neuromedin U (NMU). These data may suggest regulatory functions of ILC2s in Type II acquired immunity by IL-10 production. However, CD4^+^ T cells produced more IL-10 than ILC2s in our papain model mice [149]. ILC2-specific IL-10 deletion is required to precisely determine the physiological roles of IL-10 derived from ILC2s.

We also identified that a small population of IL-10-producing ILC2s in the bronchoalveolar space expressed inhibitory receptor TIGIT and exhibited hyporesponsiveness, as determined by low T_H_2 cytokine production and low proliferation during severe airway allergy (Figure 3b) [149]. TIGIT^+^IL-10^+^ ILC2s also expressed checkpoint inhibitory receptor PD-1. Transcriptome analysis revealed the similarity between TIGIT^+^IL-10^+^ ILC2s in the bronchoalveolar space and CD8^+^ T cells exhausted by chronic LCMV infection. Therefore, we hypothetically designate the hyporesponsive ILC2s expressing checkpoint inhibitory receptors as “exhausted-like” ILC2s [149,151]. TIGIT^+^IL-10^+^ ILC2s were preferentially localized at the site of severe inflammation, such as bronchoalveolar space, and rarely in the lung parenchyma. Emergence of TIGIT^+^IL-10^+^ ILC2s was enhanced by the absence of transcription factor Runx proteins [149]. Interestingly, deficiency of Cbfβ, a binding partner of all Runx proteins, or both Runx1 and Runx3 in ILC2s augmented the emergence of TIGIT^+^IL-10^+^ ILC2s during allergic inflammation in vivo and vitro, resulting in reduced allergic inflammation accompanied by reduced eosinophils and GATA-3^+^ CD4^+^ T cells. Since Cbfβ/Runx complexes are associated with gene loci such as IL-5, IL-13, IL-10, and TIGIT in ILC2s stimulated by IL-33, Runx proteins may directly suppress the phenotype of “exhausted-like” ILC2s in severe or repeated allergic inflammation [149]. Collectively, our data suggest that the induction of the “exhausted-like” phenomenon in ILC2s ameliorates allergic inflammation due to reduced eosinophil recruitment and T_H_2 skewing.

However, the physiological relevance of “exhausted-like” ILC2s remains elusive because the specific deletion of TIGIT^+^IL-10^+^ ILC2s is quite difficult. As TIGIT^+^IL-10^+^ ILC2s express some inhibitory molecules, deletion of a single molecule may not be enough to release ILC2s from their dysfunctional state. Hyporesponsive ILC2s are generally rare in chronic allergy induced by continuous papain nasal treatment [149]. In addition, we do not know whether such dysfunctional ILC2s really exist in patients with chronic allergies. Once ILC2s are activated by IL-33 or allergens in vivo, such activated ILC2s live long and secrete more T_H_2 cytokines upon a second challenge of a different antigen. The question remains of how the fate of activated ILC2s is determined toward long-lived trained response or exhausted-like low reactivity.

## 4. ILC3s and Acquired Immunity

Lti cells are necessary for the formation of secondary lymphoid tissue where T cells and B cells encounter antigen-presenting cells. In the absence of Lti cells, lymph nodes, Peyer’s patches, and the splenic white pulp do not form in mice [8]. Therefore, Lti cells are obviously required for acquired immunity. Aside from secondary lymphoid organogenesis by Lti cells, we here review molecular and cellular pathways that link ILC3s and acquired immunity, mainly in the intestine.

### 4.1. Innate Production of IL-17 and IL-22 by ILC3s

IL-17 and IL-22 are signature cytokines for Type III immune responses that protect the host from the intrusion of bacterial and fungal extracellular pathogens, including *Citrobacter rodentium*, *Salmonella typhimurium*, and *Candida albicans* (Figure 4a) [150,152]. Both IL-17 and IL-22 target epithelial cells to induce antimicrobial peptides and CXC chemokines that attract neutrophils to the site of inflammation for the engulfment of pathogens [153,154,155]. IL-17 is recognized as an inflammatory cytokine because IL-17, together with TNF-α, IL-6, and IL-1, promotes neutrophil activation and is involved in autoimmunity pathogenesis [152,156,157,158,159]. In contrast, IL-22 is a protective cytokine because the cytokine is indispensable for tissue protection and regenerative process of the epithelium [150]. IL-22 mediates intestinal goblet-cell hyperplasia, mucus production, and epithelial fucosylation, which directly inhibits the access of harmful pathogens to intestinal epithelial cells [160,161]. IL-22 also promotes epithelial-cell proliferation to maintain mucosal integrity during bacterial infection [150,162]. IL-22 administration augments epithelial regeneration through the recovery of intestinal stem cells that are damaged in GVHD-model mice [162]. However, excessive IL-22 can deteriorate CD40-induced colitis by enhancing neutrophil recruitment [163]. Collectively, IL-17 and IL-22 are essential for innate membrane barrier and mucosal immunity against extracellular pathogens.

In the search for IL-17- or IL-22-producing cells, ILC3s have been identified as an innate source of IL-17 and IL-22 in the intestine, Peyer’s patches, mesenteric lymph nodes, meninges, and skin [5,26,35,164,165]. Data on IL-22 reporter mice suggested that ILC3s are major IL-22 producers, especially in the intestine [166]. Deletion of ILC3s is associated with an impaired gut mucosal barrier and inflammatory responses in the intestine [167]. When IL-22-producing ILCs are deleted in Rag1^–/–^ mice, *Alcaligenes* species disseminate through the impaired mucosal barrier and cause systemic inflammation. ILC3 activity is modulated by intestinal mononuclear phagocytes (MNPs) armed with TLRs that recognize PAMPs, such as the bacterial cell wall and flagellin [168,169]. After receiving danger signals, human and mouse MNPs secrete IL-23 and IL-1β, both of which activate ILC3s, T_H_17 cells, and T_H_22 cells [169,170,171]. In turn, ILC3s secrete GM-CSF for MNPs to produce IL-1β in humans and mice [168]. ILC3s are also adjacent to nerve endings and are stimulated by neurotropic factors, neuropeptides, and neurotransmitters. ILC3s in mouse intestine express neuroregulatory receptor RET, and are activated by glial-derived neurotrophic-factor family ligands secreted from glial cells [172]. Signaling through TLR2 and TLR4 induces the expression of the neurotrophic factor in the glial cells, suggesting that enteric glial cells sense PAMPs and initiate protective immune responses through ILC3 activation [172]. Neuropeptide VIP also enhances ILC2 and ILC3 activity through its cognate receptor VIPR2, expressed on murine enteric ILC2s and ILC3s [173,174]. Interestingly, food intake elicits VIP secretion from intestinal neurons leading to IL-5 production by ILC2s and IL-22 production by ILC3s in mice. ILC3s can be stimulated by acetylcholine from the vagal nerve, and produce host-protective lipid mediator PCTR1 (16Rglutathionyl, 17S-hydroxy-4Z,7Z,10Z,12E,14E,19Z-docosahexaenoic acid) [175]. ILC3s are also subjected to direct regulation by microbial and dietary metabolites. Butylate, a short-chain fatty acid, is generated by microbiota and has a suppressive effect on ILC3s through GPR109a-mediated signaling [176]. Maternal retinoic acid, a vitamin A metabolite, promotes LTi-cell differentiation in embryos [177]. These data indicated that environmental cues directly or indirectly influence ILC3 activity.

### 4.2. ILC3s Facilitate Acquired Immune Responses

From the perspective of the innate helper function of ILCs, ILC3s should augment T_H_17 and T_H_22-cell activity upon pathogenic-bacterium infection, like NK cells and ILC2s enhance T_H_1- and T_H_2-cell differentiation, respectively. However, studies of *Citrobacter rodentium* infection revealed that ILC3s may play a limited role in facilitating T_H_17- and T_H_22-cell responses against bacterial infection. In mice where ILC1s and NCR^+^ ILC3s are lacking, we and others observed the aggravation of *Citrobacter rodentium* infection around day 10 post infection [22,26], whereas this adverse effect disappears in the late phase of infection [178]. This may be due to pre-existing Th17 and/or Th22 cells in the intestine, which may offer early acquired immune responses against *Citrobacter rodentium* and compensate for the loss of ILC3s. However, another report showed that ILC3s supported T_H_17-cell differentiation induced by segmented filamentous bacteria (SFB), a commensal bacterium [179] (Figure 4a). Intestinal epithelial cells are stimulated by IL-22 from ILC3s and secrete serum amyloid A, which accelerates IL-17 expression in T_H_17 cells. However, in AHR-deficient mice, a reduction in IL-22-producing ILC3 numbers results in the increased colonization of SFB and high T_H_17 differentiation [180]. It appears that ILC3s, T_H_17 cells, and the microbiome are mutually associated to establish mucosal immunity. Nonetheless, loss of ILC3s does not seem to significantly impact T_H_17- or T_H_22-cell activation against *Citrobacter rodentium,* which is commonly used to study Type III immune responses.

In contrast, immune communication between ILC3s and T_H_1 cells was clarified in the colitis model, where immunocompetent mice were treated with dextran sulfate sodium (DSS) [25] (Figure 4a). Disruption of intestinal epithelial cells caused by DSS permits the entry of bacteria and their products into lamina propria, leading to inflammation. In their model, CX3CR1^+^ MNPs secreted TL1A in response to inflammatory bowel disease-related bacteria and activated MHC Class II^+^ ILC3s through DR3, a receptor for TL1A. Costimulatory molecule CD40L is induced on the activated MHC Class II^+^ ILC3s, which then facilitate differentiation of T_H_1, but not Treg or T_H_17 cells, by antigen presentation [25]. IFN-γ from ILC3s can be a deteriorating factor in inflammatory bowel diseases. Phenotypical conversion of NCR^+^ ILC3s to IFN-γ-producing ILC1s is prominent in intestine from Crohn’s disease patients [181,182]. IL-12 signaling confers NCR^+^ ILC3s the ability to produce IFN-γ, though IL-23, IL-1β, and RA can reverse phenotypical change [182] (Figure 4a). In mice, the IFN-γ-producing ex-ILC3s deteriorate innate colitis induced by anti-CD40 antibody injection when they are transferred to *RAG2*^-/-^*Il2rgc*^-/-^ mice [183]. Collectively, ILC3s can contribute to colitis pathogenesis by T_H_1 skewing resulting from direct antigen presentation and, presumably, by acquisition of an IFN-γ-producing ability.

Emerging evidence suggests that ILC3s can modulate humoral immunity (Figure 4a). ILC3s in gut lamina propria support T-cell-dependent IgA production by producing soluble lymphotoxin a (sLTa3) that attracts T cells and IgM^+^ B cells to the gut [184]. In contrast, membrane-bound lymphotoxin b (mLTab2) expressed on ILC3s induces iNOS^+^DCs, which are a prerequisite for IgA production, resulting in T-cell-independent IgA production by B cells. In human and mouse lymph nodes, ILC3s are present at the marginal sinus and interfollicular spaces near the marginal zone [14,185,186]. In humans, ILC3s provide the B-cell activation factor (BAFF), CD40L, and Notch ligand Delta-like 1 (DLL1) for marginal-zone B cells to produce innate IgM [186]. Therefore, ILC3s are critical for T-cell-dependent/-independent B-cell responses in the intestine and secondary lymphoid tissue.

Tertiary lymphoid structures (TLSs) are ectopic lymphoid structures which can be found in tissues with chronic inflammation. ILC3s are associated with TLS formation in some pathological conditions [187]. ILC3s accumulate in the lung and protect hosts by formation of lymphoid structures in granuloma following *Mycobacterium tuberculosis* (Mtb) infection in mice [188]. CXCL13 is induced in the Mtb-infected lung and recruits circulating CXCR5^+^ ILC3s to the lung. Human NCR^+^ ILC3s are localized at the edge of TLSs in non-small-cell-lung cancer (NSCLC) [189]. Frequencies of human NCR^+^ ILC3s in NSCLC are correlated with good outcome. NCR^+^ ILC3s interact with lung tumor cells and tumor-associated fibroblasts through NKp44, leading to production of TNF-α and LTαβ both of which are critical for Lti function. Despite these observations, Lti cells and ILC3s are not required for formation of TLSs in the lung repeatedly treated with LPS [190]. Therefore, types of inflammation may differentially affect Lti functions of ILC3s.

### 4.3. Regulatory Functions of ILC3s

Contrary to the discussion above, ILC3s can directly inhibit CD4^+^ T-cell activation (Figure 4b). As CCR6^+^ ILC3s express MHC Class II but not costimulatory molecules in the intestine with commensal bacteria, ILC3s elicit the cell death of microbiota-specific CD4^+^ T cells after improper antigen presentation at the intestine and mesenteric lymph nodes [24]. Deletion of MHC Class II in ILC3s causes colitis due to microbiota-specific CD4^+^ T cells. Likewise, MHC Class II^+^ ILC3s reduce the number of T_H_17 cells induced by SFB [191]. In parallel with this, MHC Class II expression on enteric ILC3s is reduced in pediatric Crohn’s disease patients [24]. A recent report identified that ILC3s in peripheral lymph nodes express an autoimmune regulator (Aire) that induces promiscuous tissue antigen presentation on medullary thymic epithelial cells for T-cell tolerance [192]. Aire^+^ ILC3s express costimulatory molecules and directly present antigens to CD4^+^ T cells. However, the physiological function of Aire^+^ ILC3s remains to be determined. ILC3s can also modulate Treg-cell fitness in human and mouse intestines (Figure 4b). Intestinal ILC3s predominantly secrete IL-2 that promotes Treg expansion for immune tolerance to dietary antigen [193]. IL-2 from ILCs, but not T cells, is crucial for the homeostasis of Treg cells in the intestine. In line with these observations, IL-2^+^ILC3s are reduced in the intestine of patients with Crohn’s disease. A recent study revealed that OX40L on murine ILC3s provoked a proliferation signal in Treg cells via its receptor, OX40 [194]. Thus, ILC3s maintain gut homeostasis by suppressing commensal bacterium-specific CD4^+^ T cells and promoting Treg-cell expansion in the intestine.

ILC3s negatively regulate T-cell-dependent IgA production through controlling Tfh cells (Figure 4b). Lti-like cells deficient for MHC Class II lead to increased numbers of Tfh cells and germinal center B cells with increasing class switching to IgG1 and IgA in colon mesenteric lymph nodes [195]. Similar phenomena are not observed in Peyer’s patches and small-intestine mesenteric lymph nodes, where polyclonal IgA is predominantly produced in a T-cell-independent manner. In mice infected with *Citrobacter rodentium*, MHC Class II^+^ ILC3s suppress Tfh-cell proliferation and IgA production specific to the bacteria [195]. Therefore, ILC3 interaction with Tfh cells is critical for suppression of T-cell-dependent IgA responses toward mucosal bacteria.

## 5. Conclusions

Since ILCs and helper CD4^+^ T cells are functionally related to each other, each ILC subset should have positive effects in the development of each type of immune response. However, the eventual outcome from T and B cells modulated by activated ILCs is not so simple. NK cells are recognized as innate protector cells against viral infection. This is not the case, given that activated NK cells target virus-specific CD4^+^ and CD8^+^ T cells. While ILC2s fuel allergic immune responses, especially in the acute phase, dysfunctional ILC2s may no longer promote allergies or induce T_H_2 activation in the chronic allergy. ILC3s can support T_H_1 activation for the development of colitis, though ILC3s were originally suggested as protective immune cells. Interactions between ILCs and T cells are now becoming clear. However, how humoral immunity is influenced by ILCs remains to be further elucidated. In addition, the physiological relevance of ILCs in chronic inflammation is not clear. Deep insight into the cross talk between ILCs and acquired immune cells should help us to understand the significance of ILCs in patients suffering from ILC-associated inflammation, such as inflammatory bowel diseases, psoriasis, asthma, and atopic dermatitis.

## Figures and Tables

**Figure 1 cells-09-01193-f001:**
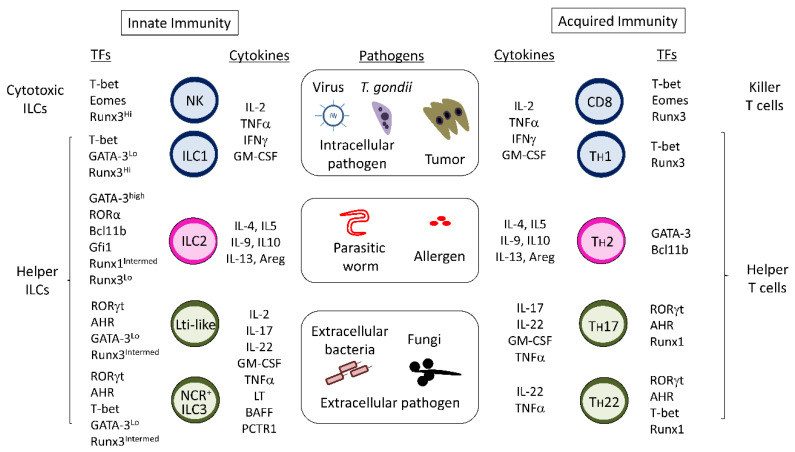
Similarities between innate lymphoid cells (ILCs) and T cells. TF: transcription factor.

**Figure 2 cells-09-01193-f002:**
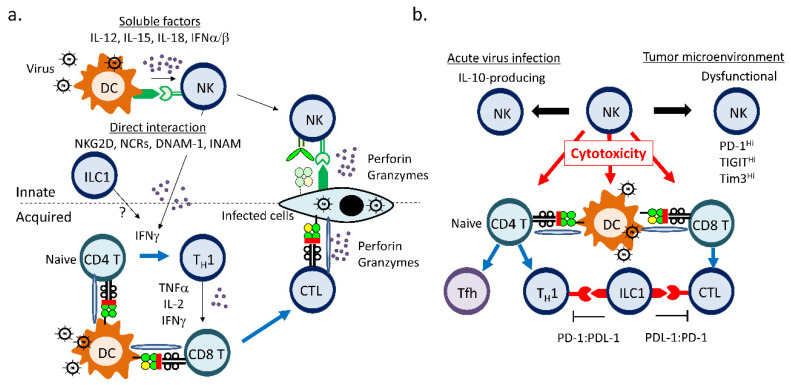
Natural-killer (NK) cells and ILC1s positively or negatively regulate acquired immunity. (**a**) NK cells enhance Type I immunity mediated by T_H_1 cells. NK cells are highly activated after mutual interaction with dendritic cells (DCs) that sense pathogen-associated pattern molecules (PAMPs) such as virus-derived RNAs and DNAs. Activated NK cells secrete IFN-γ for T_H_1 skewing. (**b**) NK cells and ILC1s suppress antivirus acquired immunity. NK cells attack CD4^+^ T cells, CD8^+^ T cells, and infected-DCs to inhibit acquired immune responses against virus infection. PDL-1 on hepatic ILC1s suppresses anti-virus CD4^+^ and CD8^+^ T cells through PD-1 on T cells. IL-10-producing NK cells and dysfunctional NK cells may be involved in suppressing cytotoxic T lymphocyte (CTL) responses.

**Figure 3 cells-09-01193-f003:**
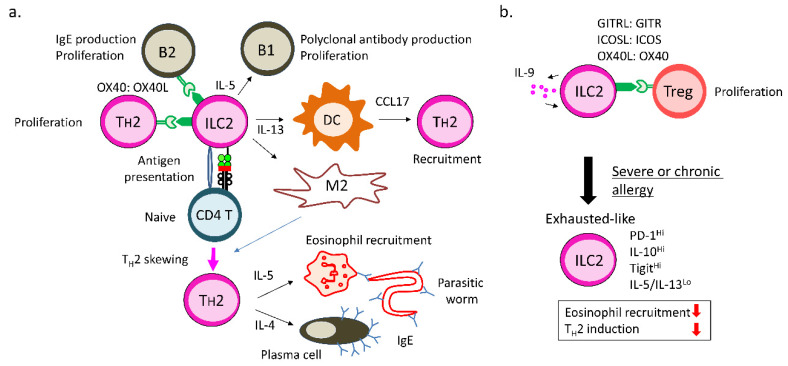
ILC2s modulate acquired immune responses. (**a**) ILC2s enhance Type II immune responses against parasitic worms. ILC2s secrete IL-5 that promotes B1-cell proliferation. OX40L/OX40 signaling is required for B2-cell proliferation and IgE production against helminth. ILC2s elicit T_H_2 polarization through recruitment of CCL17-producing DCs, induction of M2 macrophages, and direct antigen presentation. OX40L on ILC2s contributes to T_H_2 proliferation by interacting with OX40 expressed on T_H_2 cells. (**b**) Immunosuppressive function of ILC2s. IL-9 stimulates ILC2 survival and activity by autocrine and support Treg-cell expansion via GITRL/GITR, ICOSL/ICOS, and OX40L/OX40 signaling. At the site of severe or chronic allergic inflammation, ILC2s can be dysfunctional “exhausted-like” ILC2s that express checkpoint inhibitory receptors and produce immunosuppressive IL-10. Induction of exhausted-like phenomenon in ILC2s suppresses allergy pathogenesis associated with eosinophil recruitment and T_H_2 induction.

**Figure 4 cells-09-01193-f004:**
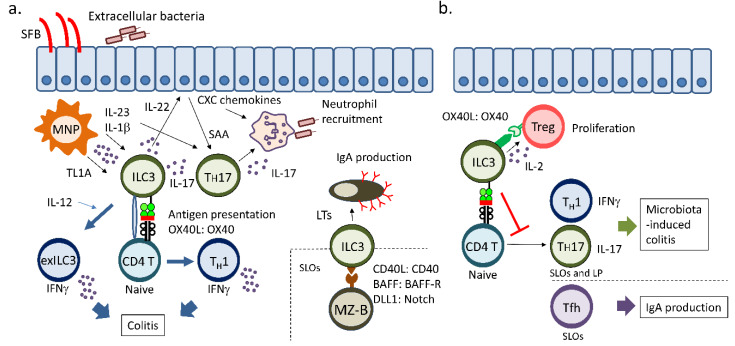
ILC3s control subsequent acquired immunity. (**a**) ILC3s augment T_H_1 and T_H_17 differentiation. Colonization of segment filamentous bacteria (SFB) enhances T_H_17 differentiation through ILC3-derived IL-22 that induces intestinal epithelial cells to secrete serum amyloid A (SAA) for T_H_17 differentiation. ILC3s present antigens to naïve CD4^+^ T cells and elicit T_H_1 differentiation in colitis. NCR^+^ ILC3s acquire an ability to produce IFN-γ in an IL-12-dependent manner. ILC3s produce lymphotoxins (LTs) that support T-cell-dependent and -independent IgA production in the intestine. At secondary lymphoid organs (SLOs), ILC3s activate marginal zone B cells (MZ-B) by direct contact. (**b**) Immunosuppressive roles of ILC3s in regulating acquired immunity. ILC3s promote Treg expansion by IL-2 and OX40/OX40L signaling. ILC3s suppress differentiation of microbiota-specific T_H_1 and T_H_17 cells by antigen presentation without costimulatory signals. Similarly, Tfh-cell differentiation in colon mesenteric lymph nodes is suppressed by ILC3s. LP: lamina propria.

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
