# Peer review of "Dichotomous Regulation of Acquired Immunity by Innate Lymphoid Cells"

_cells, 2020, doi:10.3390/cells9051193_

Round 1
Reviewer 1 Report
The review is well written and summarizes recent advances in understanding the extent to which ILCs may regulate adaptive immune responses. The author wrote a good introduction presenting the main properties of ILCs. The review includes a comprehensive description of the role of distinct ILCs populations in different types of immune response focusing on their impact on lymphocytes through a range of mechanisms. In particular, the author summarizes recent findings on the role of each ILC population as positive and negative regulator of T and B-mediated response. Finally, literature is correctly cited.
Minor concerns:
- In line 91 the author wrote “NK cells and ILC1s are innate components of Type I immunity of which the purpose is” and in line 94 wrote “ILC1s and NK cells are a phenotypically very similar population”. Please write both phrases in a clear form.
- In line 99 please replace “hepatic ILC1s seems…”
-In line 106, LFA-1 is listed as activating receptor but it is an adhesion molecule.
-In line 285 please correct “we tentatively name…”
Author Response
I thank the reviewer#1 for the favorable remarks and comments that help to improve my manuscript. I have addressed these comments (in italics), as noted in our point-by-point replies below. In the manuscript itself, you can find changes with "Track Changes" function in Microsoft Word.
1) In line 91 the author wrote “NK cells and ILC1s are innate components of Type I immunity of which the purpose is” and in line 94 wrote “ILC1s and NK cells are a phenotypically very similar population”. Please write both phrases in a clear form.
2) In line 99 please replace “hepatic ILC1s seems…”
These sentences were deleted or rephrased as follows (line 96-98):
NK cells and ILC1s are innate components of Type I immunity which provides protective responses against tumor cells or intracellular microbes, such as viruses, bacteria, and protozoa.
3) In line 106, LFA-1 is listed as activating receptor but it is an adhesion molecule.
LFA-1 was deleted.
4) In line 285 please correct “we tentatively name…”
The sentence was rephrased as follows:
We hypothetically designate the hyporesponsive ILC2s expressing checkpoint inhibitory receptors as “exhausted-like” ILC2s.
Reviewer 2 Report
The author of this article titled "The Interplay between Innate Lymphoid Cells and Acquired Immunity" reviews recent advances in our understanding of the interactions between ILCs and acquired immunity. The author treads the same ground as the authors of many recently published reviews, including Withers, 2016 Immunology, 149(2):123-130; Sonnenberg and Hepworth, 2019 Nature Rev. Immunol., 19:599–613; and Hepworth and Sonnenberg, 2014 Current Opinion in Immunol., 25:75-82. There is little new in this offering from Ebihara.
Additionally, there is no discernible structure to this review. The papers reporting studies from human and mouse are often interleaved as are the pathological conditions being investigated (e.g. cancer and viral infections). Therefore, there is no significant clarity brought to this field.
It would be also important to flag the shortfalls in the existing studies and identify where improvements are needed.
Last but not least, the article requires extensive editing of English language and style!
Author Response
I thank the reviewer#2 for the favorable remarks and comments that help to improve my manuscript. I have addressed these comments (in italics), as noted in our point-by-point replies below. In the manuscript itself, you can find changes with "Track Changes" function in Microsoft Word.
1) The author of this article titled "The Interplay between Innate Lymphoid Cells and Acquired Immunity" reviews recent advances in our understanding of the interactions between ILCs and acquired immunity. The author treads the same ground as the authors of many recently published reviews, including Withers, 2016 Immunology, 149(2):123-130; Sonnenberg and Hepworth, 2019 Nature Rev. Immunol., 19:599–613; and Hepworth and Sonnenberg, 2014 Current Opinion in Immunol., 25:75-82. There is little new in this offering from Ebihara.
This comment is right. However, I intend to clearly state the functional dichotomy in ILCs which can be positive or negative regulators of acquired immunity. I added sentences to clarify the aim of this review (line 85-91). Title was also changed to “Dichotomous Regulation of Acquired Immunity by Innate Lymphoid Cells”.
2) Additionally, there is no discernible structure to this review. The papers reporting studies from human and mouse are often interleaved as are the pathological conditions being investigated (e.g. cancer and viral infections). Therefore, there is no significant clarity brought to this field.
Descriptive structure especially on NK cells and ILC1s was totally edited. Regulatory functions of human NK cells were separately described in line 184-196 as suggested by reviewer#2 and #3.
3) It would be also important to flag the shortfalls in the existing studies and identify where improvements are needed.
The relevant descriptions can be found in line 134-136, 142-143, 157-160, 212-221, 334-335, 354-362, and 529-533. I would like to add more. However, I reviewed quite broad research area especially for researches who do not know about ILCs and NK cells. So that, the paper should be straight-forward and simple. I abandoned to discuss each thematic research area more in details.
4) Last but not least, the article requires extensive editing of English language and style!
I improved my manuscript as much as I can. The paper was proofread by a native English speaker.
Reviewer 3 Report
In this manuscript, the author reviews the crosstalk between ILCs and acquired immunity. The paper is well written, well organized in sections and very well detailed for each ILC subset. The wide array of interactions between ILCs and T cells the author covers are based on a very thorough bibliographical search. I really think the review will be of great value for the scientific community working on the pathophysiology of ILCs.
I only have very minor comments:
- The author refers to group 1 ILCs to categorize NK cells and ILC1s and to three subpopulations of ILCs. However, the new nomenclature approved by the International Union of Immunological Societies presented by Vivier and Spits (Cell 2018) proposes to classify ILCs into five subsets (NK cells, ILC1s, ILC2s, ILC3s and LTi cells), based on their development. Can the author adhere to this nomenclature as well?
- I the ILC2 part (line 216) the author describes the regulation of ILC2 activation by neuropeptides, lipid mediators or glucocorticoids. Can the author describe better how these factors actually regulate type II immune responses in a physiological or pathological context?
- The author describes emerging evidence suggesting that ILC3s can modulate humoral immunity. I would suggest to add here some reports regarding the role of ILC3s on tertiary lymphoid tissue formation in some pathological contexts.
Author Response
I thank the reviewer#3 for the favorable remarks and comments that help to improve my manuscript. I have addressed these comments (in italics), as noted in our point-by-point replies below. In the manuscript itself, you can find changes with "Track Changes" function in Microsoft Word.
1) The author refers to group 1 ILCs to categorize NK cells and ILC1s and to three subpopulations of ILCs. However, the new nomenclature approved by the International Union of Immunological Societies presented by Vivier and Spits (Cell 2018) proposes to classify ILCs into five subsets (NK cells, ILC1s, ILC2s, ILC3s and LTi cells), based on their development. Can the author adhere to this nomenclature as well?
I followed the new nomenclature. The descriptions, wording, and figure 1 were properly edited.
2) In the ILC2 part (line 216) the author describes the regulation of ILC2 activation by neuropeptides, lipid mediators or glucocorticoids. Can the author describe better how these factors actually regulate type II immune responses in a physiological or pathological context?
I described the effects of neuropeptides and lipid mediators on ILC2s more in details (line 267-276).
3) The author describes emerging evidence suggesting that ILC3s can modulate humoral immunity. I would suggest to add here some reports regarding the role of ILC3s on tertiary lymphoid tissue formation in some pathological contexts.
I discussed functions of ILC3s in formation of tertiary lymphoid structure in line 468-478.
Reviewer 4 Report
This is a thorough review of ILC biology focused on the ability of these cells to influence adaptive immune responses. Dr. Ebihara did a nice job of covering the major thematic areas for each ILC type and the figures are wonderful. The writing is a little choppy and repetitive, but very readable.
One major omission that struck me was the body of work on human NK cell regulatory function. Papers from Peppa et al 2013 (HBV), Bradley et al 2018 (HIV), Nielsen et al. 2012, Grutza et al 2020, Mooney et al 2020, and Cerboni et al 2007 represent a set (not inclusive) of studies done with human NK cells that should be included. There are probably some human studies with other types of ILCs; one from Andrea Cerutti about ILC1 making BAFF comes to mind but is probably not the only one.
A second area to consider is the wealth of data around NKG2A and Qa1/HLA-E in control of NK cell regulation of T cells. NKG2A is mentioned, but there are works from Harvey Cantor, Phillip Lang, and Ray Welsh that should be discussed around this point. The new Grutza paper mentioned above is a nice counterpoint about memory/adaptive NK cells that use NKG2C and HLA-E for regulation (positive signal).
A few other important papers on the NK cell regulatory front include the 2011 Soderquest paper, 2018 Rydyznski (non-viral), 2018 Blass paper, Nakayama et al 2011, and 2004 Zingoni (OX40 interactions between NK and CD4, there is one on CD40 as well).
The section on IL-10 production by NK cells misses a few papers from Markus Mohrs and Sara Hamilton and Laurel Lenz, etc.
The section about IFN-g around line 133 could be enhanced by a recent vaccine study from GSK where adjuvant induced NK cell gamma enhances vaccine responses (Published in npj vaccines).
Author Response
I thank the reviewer#4 for the favorable remarks and comments that help to improve my manuscript. I have addressed these comments (in italics), as noted in our point-by-point replies below. In the manuscript itself, you can find changes with "Track Changes" function in Microsoft Word.
1) One major omission that struck me was the body of work on human NK cell regulatory function. Papers from Peppa et al 2013 (HBV), Bradley et al 2018 (HIV), Nielsen et al. 2012, Grutza et al 2020, Mooney et al 2020, and Cerboni et al 2007 represent a set (not inclusive) of studies done with human NK cells that should be included. There are probably some human studies with other types of ILCs; one from Andrea Cerutti about ILC1 making BAFF comes to mind but is probably not the only one.
Human NK cell data are discussed in line 184-196 as the reviewer #3 commented. The papers were properly cited in the section of “Regulatory functions of Group 1 ILCs”. ILC3s producing BAFF were discussed in 464-465.
2)A second area to consider is the wealth of data around NKG2A and Qa1/HLA-E in control of NK cell regulation of T cells. NKG2A is mentioned, but there are works from Harvey Cantor, Phillip Lang, and Ray Welsh that should be discussed around this point. The new Grutza paper mentioned above is a nice counterpoint about memory/adaptive NK cells that use NKG2C and HLA-E for regulation (positive signal).
I discussed the function of NKG2A in line 106-108 and 212-221. The papers were properly cited.
3) A few other important papers on the NK cell regulatory front include the 2011 Soderquest paper, 2018 Rydyznski (non-viral), 2018 Blass paper, Nakayama et al 2011, and 2004 Zingoni (OX40 interactions between NK and CD4, there is one on CD40 as well).
The papers were properly discussed or cited in line 131-136 and 162-183.
4) The section on IL-10 production by NK cells misses a few papers from Markus Mohrs and Sara Hamilton and Laurel Lenz, etc.
The papers were properly discussed in line 222-237.
5)The section about IFN-g around line 133 could be enhanced by a recent vaccine study from GSK where adjuvant induced NK cell gamma enhances vaccine responses (Published in npj vaccines).
The paper was discussed in line 128-136.
Round 2
Reviewer 2 Report
No further comments.